Genome-wide identification, subcellular localization, and expression analysis of the phosphatidyl ethanolamine-binding protein family reveals the candidates involved in flowering and yield regulation of Tartary buckwheat (Fagopyrum tataricum)

Nie Mengping 1
Li Li 1
He Cailin 1
Lu Jing 1
Guo Huihui 1
Li Xiao’an 1
Jiang Mi 2
Zhan Ruiling 2
Sun Wenjun 1
Yin Junjie 3
Wu Qi 1 jerviswuqi@126.com
1 Key Laboratory of Coarse Cereal Processing, Ministry of Agriculture and Rural Affairs, Sichuan Engineering & Technology Research Center of Coarse Cereal Industralization, College of Food and Biological Engineering, Chengdu University , Chengdu, Sichuan , China
2 Key Laboratory of Wheat Crop Research in Ganzi Academy of Agricultural Sciences, Ganzi Academy of Agricultural Sciences , Ganzi, Sichuan , China
3 State Key Laboratory of Crop Gene Exploration and Utilization in Southwest China, Sichuan Agricultural University , Chengdu, Sichuan , China
Gelfand Mikhail
Electronic publication date: 2024 Mar 26
Publication date: 2024
Volume: 12
Electronic Location ID: e17183
Received 2023 Nov 28; Accepted 2024 Mar 11
Copyright: © 2024 Nie et al.
Copyright year: 2024
Copyright holder: Nie et al.
License: This is an open access article distributed under the terms of the Creative Commons Attribution License, which permits unrestricted use, distribution, reproduction and adaptation in any medium and for any purpose provided that it is properly attributed. For attribution, the original author(s), title, publication source (PeerJ) and either DOI or URL of the article must be cited.
License URL: https://creativecommons.org/licenses/by/4.0/

Keywords: FT/TFL, Flowering, Yield, Tartary buckwheat

Funding: Natural Science Foundation of Sichuan Province 2022NSFSC1773, 2022NSFSC1725, 2023ZHCG0091, 2023NSFSC1177 National Natural Sciences Foundation 32301850 Open Project Program of State Key Laboratory of Crop Gene Exploration and Utilization SKL-KF202302 China Agriculture Research System CARS07-B-1 Ganzi Science and Technology Program 220015 This work was supported by the Natural Science Foundation of Sichuan Province (Grant 2022NSFSC1773, 2022NSFSC1725, 2023ZHCG0091, 2023NSFSC1177), the National Natural Sciences Foundation of China (No. 32301850), the Open Project Program of State Key Laboratory of Crop Gene Exploration and Utilization in Southwest China (Grant No. SKL-KF202302), the China Agriculture Research System (Grant CARS07-B-1) and the Ganzi Science and Technology Program (Grant 220015). The funders had no role in study design, data collection and analysis, decision to publish, or preparation of the manuscript.

==============================
Background

PEBP (phosphatidyl ethanolamine-binding protein) is widely found in eukaryotes including plants, animals and microorganisms. In plants, the PEBP family plays vital roles in regulating flowering time and morphogenesis and is highly associated to agronomic traits and yields of crops, which has been identified and characterized in many plant species but not well studied in Tartary buckwheat (Fagopyrum tataricum Gaertn.), an important coarse food grain with medicinal value.

Methods

Genome-wide analysis of FtPEBP gene family members in Tartary buckwheat was performed using bioinformatic tools. Subcellular localization analysis was performed by confocal microscopy. The expression levels of these genes in leaf and inflorescence samples were analyzed using qRT-PCR.

Results

Fourteen Fagopyrum tataricum PEBP (FtPEBP) genes were identified and divided into three sub-clades according to their phylogenetic relationships. Subcellular localization analysis of the FtPEBP proteins in tobacco leaves indicated that FT- and TFL-GFP fusion proteins were localized in both the nucleus and cytoplasm. Gene structure analysis showed that most FtPEBP genes contain four exons and three introns. FtPEBP genes are unevenly distributed in Tartary buckwheat chromosomes. Three tandem repeats were found among FtFT5/FtFT6, FtMFT1/FtMFT2 and FtTFL4/FtTFL5. Five orthologous gene pairs were detected between F. tataricum and F. esculentum. Seven light-responsive, nine hormone-related and four stress-responsive elements were detected in FtPEBPs promoters. We used real-time PCR to investigate the expression levels of FtPEBPs among two flowering-type cultivars at floral transition time. We found FtFT1/FtFT3 were highly expressed in leaf and young inflorescence of early-flowering type, whereas they were expressed at very low levels in late-flowering type cultivars. Thus, we deduced that FtFT1/FtFT3 may be positive regulators for flowering and yield of Tartary buckwheat. These results lay an important foundation for further studies on the functions of FtPEBP genes which may be utilized for yield improvement.

Introduction

Tartary buckwheat (Fagopyrum tataricum Gaertn.) is an important traditional coarse cereal with a long cultivation history in southwest China (Wen et al., 2021). It was the main food for the minorities in the marginal areas of Sichuan, Guizhou and Yunnan provinces (Wu et al., 2017). Because Tartary buckwheat seeds are abundant in flavonoids, active peptides and minerals (Ma et al., 2019; Huang et al., 2016), they are usually used as food and medicine (Liu et al., 2021). Recently, with the increasing recognition of Tartary buckwheat’s nutritional and medicinal value, the global demand is growing rapidly. However, compared with the staple food crops (rice, wheat), the yield of Tartary buckwheat still has a large room for improvement. Thus, expanding the planting area and improving the yield of Tartary buckwheat is necessary. Flowering regulation has been reported to influence the inflorescence morphology and regional adaptation of plants and is closely related to crop yield (Song et al., 2015). Some Tartary buckwheats with long growth periods usually could not survive the extremely hot or cold weather to generate seeds. Rice Tartary buckwheat is a specific cultivar that originated around Himalaya, and is favored by many people because of the easy-to-dehull properties of the seeds (Li et al., 2020). As its floral transition process is usually hindered by temperature or light conditions in low-altitude planting area, rice Tartary buckwheat has a very long vegetative phase in which inflorescence cannot elongate, resulting in very low florets production. Thus, the planting of rice Tartary buckwheat is narrowed to higher altitude area with lower temperature. Therefore, exploration and utilization of flowering and inflorescence-related genes are essential to expanding the planting region and increasing the yield of long-period Tartary buckwheat.

Flowering is an essential process for the transition from vegetative to reproductive growth (Song et al., 2015) and is influenced by internal and environmental factors (Hemming, Peacock & Trevaskis, 2008). The phosphatidyl ethanolamine-binding protein (PEBP) gene family is widespread in many species, including bacteria, animals, and plants (Karlgren et al., 2011). The PEBP family members are tightly associated with plant growth and development. Many PEBP family members have been identified in various plants, such as Arabidopsis (Arabidopsis thaliana) (Wigge et al., 2005; Fryxell, 1996), rice (Oryza sativa) (Tamaki et al., 2007; Zhao et al., 2022), soybean (Glycine max) (Wang et al., 2015; Chengming Fan et al., 2014), maize (Zea mays) (Meng, Muszynski & Danilevskaya, 2011; Danilevskaya et al., 2008) and potato (Solanum tuberosum) (Navarro et al., 2011; Zhang et al., 2022). Three sub-clades were classified according to the structure and function in plants: FT-like, TFL1-like and MFT-like subgroups (Karlgren et al., 2011). FLOWERING LOCUS T (FT) encodes for florigen protein that moves through the phloem from leaves to the shoot apical meristem (SAM) to activate flowering (Corbesier et al., 2007; Huang et al., 2005). In contrast, TERMINAL FLOWER 1 (TFL1) is a flowering repressor (Wickland & Hanzawa, 2015; Shannon & Meeks-Wagner, 1991). MOTHER OF FT AND TFL1 (MFT) is homologous to both FT and TFL1 and constitutive expression of MFT resulted in slightly earlier flowering under long days (Chen et al., 2018; Yoo et al., 2004). Besides, MFT plays vital role in seed germination and development, which promotes embryo growth through a negative feedback loop in the ABA signaling pathway (Xi et al., 2010).

To date, many studies have proved the roles of the PEBP genes in agronomic trait regulation. When Hd3a was suppressed, the transgenic plants showed a later flowering time and a reduction in the number of branches compared to the wild-type (WT) plants (Tsuji et al., 2015). Overexpression of RCN1 or RCN2, rice TFL1/CEN homologs, caused a delayed transition to the reproductive phase and displayed a more branched, denser panicle morphology (Nakagawa, Shimamoto & Kyozuka, 2002). The wheat TaTFL1-5 mutation reduced the tiller numbers per plant during the vegetative period and decreased the number of effective tillers and spikelets at the maturity stage (Sun et al., 2023a). Overexpression of HbMFT1 resulted in delayed seed germination, seeding growth, and flowering in transgenic Arabidopsis (Bi et al., 2016). The maize plants ectopic expressing ZCN8 had earlier flowering times (Meng, Muszynski & Danilevskaya, 2011; Danilevskaya et al., 2008). Yet, some PEBP genes within the same subfamily may have differing roles. In soybean, GmFT1a is a flowering inhibitor (Liu et al., 2018; Jiang et al., 2019). GmFT4, another homolog of FT, also acts as a flowering repressor (Zhai et al., 2014). Those two genes have contrasting roles to the other flowering promoters GmFT2a/5a (Nan et al., 2014). In addition to flowering controlling, FT/TFL1 is also involved in the development of plant organs. In transgenic onions (Allium cepa L.), AcFT1 promotes bulb formation, whereas AcFT4 prevents the up-regulation of AcFT1 and inhibits bulb formation (Lee et al., 2013; Rashid, Cheng & Thomas, 2019; Manoharan et al., 2016). Overexpression of StSP6A induces rapid tuberization and increases tuber yield, while up-regulation of StSP6A could inhibit bud development (Park et al., 2022; Navarro et al., 2011).

These studies provide a deep understanding of the functions of plant PEBP members, but the function of the Tartary buckwheat PEBP gene family is still unknown. In this study, based on the published genome sequence of Tartary buckwheat, we identified fourteen PEBP family genes in the genome. Then, we analyzed their phylogenetic relationships, gene structures, conserved motifs, chromosome location, and duplication events. We further analyzed the expression levels of PEBP genes two flowering-type cultivars and identified the candidate FT genes for buckwheat flowering. This study helps understand the functions of PEBP members and provides potential candidates for Tartary buckwheat breeding.

Materials and Methods

Identification of PEBP family genes in Tartary buckwheat

The genome sequences of Tartary buckwheat (Fagopyrum tataricum) and common buckwheat (Fagopyrum esculentum) were obtained from the Tartary buckwheat Genome Project (TBGP; https://www.mbkbase.org/Pinku1/) and the Chinese National Genomics Data Center database (https://bigd.big.ac.cn/) under the BioProject accession numbers PRJCA009237 (He et al., 2023), respectively. The protein sequences of Arabidopsis (A. thaliana) and rice (Oryza sativa) were downloaded from Phytozome V13 (https://phytozome-next.jgi.doe.gov). Two programs were used to identify PEBP family genes in the Tartary buckwheat genome. First, the sequences of six Arabidopsis PEBP proteins were used as queries to identify the candidate PEBP proteins by using the BLASTP program with E-value < 1.0e-10. Second, the Hidden Markov Model (HMM) profiles of the PEBP consensus conserved seed file (PF01161) were downloaded from the Pfam database (Mistry et al., 2020) and used as a query to screen the candidate PEBP proteins by the Simple HMM search tool on TBtools (E-value < 1.0e-10) (Wu et al., 2022; Chen et al., 2020). Then, all PEBP candidate proteins from the two parts were merged, and the NCBI-CDD (Marchler-Bauer et al., 2015) and InterPro databases (Matthias et al., 2020) were used to verify the PEBP proteins obtained previously. All the PEBP protein sequences can be found in Dataset 1. The theoretical isoelectric point (pI) and molecular weight (Mw) of PEBP proteins were predicted by the ProtParam program (https://web.expasy.org/protparam/). ProtComp 9.0 in the Softberry tool (http://www.softberry.com/) was used for PEBP subcellular location analysis.

Phylogenetic analysis

Based on multiple sequence alignment results of Tartary buckwheat, common buckwheat, Arabidopsis, and rice PEBP amino acid sequences obtained by using CLUSTALW (Thompson, Gibson & Higgins, 2002), a phylogenetic tree was constructed using MEGA 11.0 (Tamura, Stecher & Kumar, 2021) based on the Neighbor-Joining method (Liu et al., 2019) with a bootstrap value of 1,000. Evolview (http://evolgenius.info/) was used to add colorful visualization plots.

Gene structure and conserved motif, chromosomal locations analysis

Based on the genome sequences and general feature format (GFF) files, intron and exon structures and the physical location of PEBP genes on chromosomes were determined and visualized using the two programs Gene Structure View, and Gene Location Visualize in TBtools (Chen et al., 2020). Multiple Em for Motif Elicitation (MEME) program (https://meme-suite.org/meme/tools/meme) was used to identify the conserved motifs in PEBP proteins by setting the maximum motif count at eight, the minimum and maximum motif lengths at four and fifty amino acids, respectively (Bailey et al., 2009). The motif analysis results were displayed using the Gene Structure View program in TBtools (Chen et al., 2020).

Duplication and synteny analysis of PEBPs between Tartary buckwheat and other species

The Multiple Collinearity Scan toolkit (MCScanX) with the default parameters was used to analyze the gene duplication events (Wang et al., 2012). To investigate the homologous gene pairs of the PEBP gene family between Tartary buckwheat and the other species, we also used TBtools to analyze the inter-genomic collinearities (Chen et al., 2020).

Cis-acting element analysis

The upstream 2,000 bp sequences of the transcription start site of FtPEBP genes were extracted from the Tartary buckwheat genome sequences by TBtools (Chen et al., 2020). The cis-acting elements were screened and predicted using the PlantCARE database (http://bioinformatics.psb.ugent.be/webtools/plantcare/html/), and TBtools was used to visualize these promoter elements (Chen et al., 2020).

Gene expression analysis of FtPEBP genes during floral transition

To investigate the relationships between the expression levels of PEBP genes and the flowering time, two cultivars (MQ-Miqiao 1# and KQ-KQ178) with different flowering times were used. MQ, a rice Tartary buckwheat, has a long vegetative phase with low yield, and is a late-flowering cultivar (Wang et al., 2022; Wang & Campbell, 2007). Compared with MQ, KQ is an earlier flowering buckwheat. The two Tartary buckwheat seedlings were grown under natural field conditions at the experimental field of Chengdu University in Jianyang, Chengdu. The seeds were sown on March 17th, 2023, and samples were collected on May 13th. Although the flowering time of KQ is earlier than that of MQ, they were almost at the same growth stage when samples were collected, because both the true leaf numbers were about twelve. The young floral bud and the top two fully expanded leaves of 3–5 plants were harvested at 09:00 with three biological replicates, frozen in liquid nitrogen and stored at −80 °C for RNA extraction. According to the instructions, total RNA was extracted from various tissues using a Takara kit (Takara Biomedical Technology, Beijing, China). The RNA quantity and quality were measured using Scandrop (Jena, Germany). Approximately 3 µg of RNA was used for synthesizing the cDNA by using Prime Script RT reagent kit with gDNA Reaser (Trans Gene Biotech, Beijing, China), and 10-fold diluted the products for quantitative real-time PCR (qRT-PCR) analysis. Primers used (Table S1) for qRT-PCR were designed using GenScript (https://www.genscript.com/tools/real-time-pcr-taqman-primer-design-tool). The FtH3 gene was used as the reference gene (Liu et al., 2019). Three replications for each group were used for qRT-PCR analysis. qRT-PCR reactions were performed on the qTOWER3 Real-Time PCR Thermal Cycler (Jena, Germany) using THUNDERBIRD® SYBR® qPCR Mix (TOYOBO BIOTECH, Shanghai, China). Every qRT-PCR reaction (20 µL) included 10 µL of qPCR Mix, 2 µL of 50 mM primers, 2 µL of cDNA and 6 µL of ddH2O. The qRT-PCR program consists of 95 °C for 30 s, followed by 40 cycles of 95 °C for 5 s and 60 °C for 20 s. The 2−ΔΔCT method was used to determine the expression level (Livak & Schmittgen, 2001).

Subcellular localization analysis

Due to the lack of a stable genetic transformation system and effective transient expression system, Tartary buckwheat gene functions were usually studied through the heterologous expression systems in Arabidopsis thaliana (Sun et al., 2023b), and subcellular localization can be investigated via tobacco (Sun et al., 2019). We observed the subcellular locations of FtPEBP proteins transiently expressed in tobacco (Nicotiana tabacum L.) leaves. The CDS sequences of Tartary buckwheat PEBP genes were amplified by PCR, and CDS fragments were inserted into the KpnI and HindIII sites of binary vector pEZR(K)-LN to create the 35S::FtPEBP-GFP proteins. The primer sequences for CDS amplification were: FtFT1-CDS-1F: ATTCACTGAAATCCCACAAAACA, FtFT1-CDS-1R: TCCCTCTGGCAGTTGAAGTAG; FtFT3-CDS-1F: ATGGCAAGATCGAGAGATCC, FtFT3-CDS-1R: CACAGATGGATCTGGATAACG; FtTFL1-CDS-1F: ATGTCCAGACAGGTCATAGAGC, FtTFL1-CDS-1R: TCTTCTTCTAGCAGCAGTTTCC. The vectors were transformed into Agrobacterium tumefaciens strain GV3101 by thermal shock transformation. The transformed Agrobacterium was inoculated in a 50 mL YEB liquid medium containing 50 mg/L Kanamycin, and cultured at 28 °C until OD600 = 0.6–0.8. Centrifuge the cultured products for 5 min at 5,000 g to discard the supernatant, and Agrobacterium pellet was resuspended with the same volume of infiltration solution (containing 10 mM MES and 100 μM acetosyringone). The infiltration solution was injected into the back of tobacco leaves with a 1 mL syringe. After injection for 3 days, the GFP fluorescence signal was observed by confocal microscopy.

Results

Identification, phylogenetic relationship analysis of PEBPs in Tartary buckwheat

We used HMMER and BLASTP searches to identify the PEBP genes in Tartary buckwheat, and all the candidate PEBP members in the whole genome of Tartary buckwheat were detected. Based on NCBI-CDD, the fourteen candidate genes were further verified to harbor specific PBEP domain (Table 1). The PEBP proteins lengths ranged from 120 to 194 amino acids (aa), with an average length of 176 aa. FtFT1 had the longest coding sequence (CDS) length (585 bp), and the molecular weight and theoretical pI were 22,166.48 Da and 9.27, respectively. FtTFL5 had the shortest CDS length (363 bp), and the molecular weight and theoretical pI values were 13,302.07 Da and 6.5, respectively. In silico subcellular localization analysis showed that all the PEBP proteins are in the cytoplasm and nucleus. To investigate the subcellular localizations of Tartary buckwheat PEBP proteins in plant cells, we constructed three 35S::FtPEBP-GFP vectors, 35S::FtFT1-GFP, 35S::FtFT3-GFP, and 35S::FtTFL1-GFP, and transiently expressed them in tobacco leaf cells. The GFP fluorescence signals were observed by confocal microscopy. The results showed that all three PEBP-GFP fusion proteins were localized in both nucleus and cytoplasm (Fig. 1), consistent with the in silico prediction results.

Table 1 List of the 14 PEBP genes in Tartary buckwheat.

Gene ID	Gene name	Chromosome location	CDS (bp)	Protein (aa)	Molecular weight	Theoretical pl	Localization predicted	
FtPinG0006586300.01.T01	FtFT1	Ft1:41855259–41856572 (−)	585	194	22,166.48	9.27	Cytoplasm, Nucleus	
FtPinG0008575700.01.T01	FtFT2	Ft2:44911270–44913098 (−)	558	185	20,728.75	7.68	Cytoplasm, Nucleus	
FtPinG0008432800.01.T01	FtFT3	Ft1:33515476-33516649 (−)	543	180	20,290.07	8.55	Cytoplasm, Nucleus	
FtPinG0006092200.01.T01	FtFT4	Ft3:23659031–23662021 (−)	540	179	19,970.85	9.13	Cytoplasm, Nucleus	
FtPinG0008101900.01.T01	FtFT5	Ft4:49261859–49263744 (−)	459	152	16,988.4	8.86	Cytoplasm, Nucleus	
FtPinG0008102500.01.T01	FtFT6	Ft4:49197819–49198844 (−)	540	179	20,399.41	8.79	Cytoplasm, Nucleus	
FtPinG0004926700.01.T01	FtMFT1	Ft4:7195599–7197206 (−)	528	175	19,274.24	9.37	Cytoplasm, Nucleus	
FtPinG0004926900.01.T01	FtMFT2	Ft4:7186026–7187043 (−)	522	173	19,636.52	7.74	Cytoplasm, Nucleus	
FtPinG0005555500.01.T01	FtTFL1	Ft7:35942692–35943368 (+)	525	174	19,626.59	9.51	Cytoplasm, Nucleus	
FtPinG0001725900.01.T01	FtTFL2	Ft3:40104390–40105410 (−)	540	179	20,318.15	9.29	Cytoplasm, Nucleus	
FtPinG0008012600.01.T01	FtTFL3	Ft1:19936473–19944002 (+)	555	184	20,963.08	8.58	Cytoplasm, Nucleus	
FtPinG0001679100.01.T01	FtTFL4	Ft5:47941142–47942047 (−)	528	175	19,830.82	9.45	Cytoplasm, Nucleus	
FtPinG0001679500.01.T01	FtTFL5	Ft5:47896694–47919760 (−)	363	120	13,302.07	6.5	Cytoplasm, Nucleus	
FtPinG0004767400.01.T01	FtTFL6	Ft7:51297590–51298640 (−)	528	175	19,947.82	9.33	Cytoplasm, Nucleus	

Figure 1 Subcellular localization of empty vector and three PEBP-GFP proteins.

The empty vector and 35S::PEBP-GFP vectors were transformed into tobacco leaves, respectively, by using Agrobacterium tumefaciens mediated method. Three days later, the GFP fluorescence signal was observed by confocal microscopy.

MEGA 11.0 was used to perform sequence alignment. A phylogenetic tree was constructed. The tree was composed of fifty-eight PEBP-like protein sequences from four species, in which six PEBPs from A. thaliana, nineteen PEBPs from Oryza sativa, fourteen PEBPs from Fagopyrum tataricum and nineteen PEBPs from Fagopyrum esculentum (Fig. 2). According to the phylogenetic relationships, these genes were clustered into FT-like, TFL1-like, and MFT-like subfamily (Fig. 2). They were named as FtFT1–FtFT6, FtTFL1–FtTFL6 and FtMFT1–FtMFT2 which belonged to FT-like, MFT and TFL1-like subfamily, respectively (Fig. 2).

Figure 2 Phylogenetic tree of PEBPs from Fagopyrum tataricum (fourteen genes), Fagopyrum esculentum (nineteen genes), Oryza sativa (nineteen genes) and A. thaliana (six genes).

The proteins from each species are labeled with different graphics and colors (blue triangle: A. thaliana, yellow star: Oryza sativa, green circle: Fagopyrum tataricum, red check: Fagopyrum esculentum). A total of fifty-eight protein sequences were aligned using CLUSTALW in MEGA 11.0. The tree was constructed by MEGA 11.0 using the Neighbor-Joining method with a bootstrap of 1,000. Bootstrap values are shown on branches. Three subgroups were colored with different colors (MFT-like is colored in sky blue, TFL1-like is colored in purple and FT-like is colored in orange).

Gene structure, conserved motifs, and amino acid alignment analysis of FtPEBPs

Gene structure analysis showed that of the 14 genes, most FtPEBPs contained four exons and three introns, with the exception that FtTFL1 contained two exons and one intron (Fig. 3). The motifs prediction results showed that a total of eight motifs were identified in all FtPEBP proteins; motifs 1 to 5 were the most conserved motifs in all FtPEBP proteins, meaning that the structures of the FtPEBP members were highly conserved (Fig. 3). Motif six was only detected in FtTFL4 and FtTFL5. The varied motif structures may indicate the diverse roles of FtPEBP members from different subgroups.

Figure 3 The motifs and exon-intron structures of PEBP genes in Tartary buckwheat.

A total of eight conserved motifs were discovered among all PEBP genes identified by using MEME, and different motifs are showed in different colored boxes. FT-like, TFL1-like and MFT-like sub-clade genes are colored in orange, sky blue and purple. Exons, introns and UTRs of PEBP genes are represented by yellow boxes, dark lines and green boxes, respectively.

According to multiple amino acid sequence alignment results, we found that FtFT had the key amino acid residue tyrosine (Y) at the 106 site. At the same time, it was replaced by histidine (H) and tryptophan (W) in FtTFL and FtMFT, which is in consistent with other plants (Hanzawa, Money & Bradley, 2005) (Fig. S2). In addition, all FtFT proteins contained Arginine (R) at position 148, whereas FtTFL proteins contained Lysine (K) and FtMFT had Glutamic acid (E). Thus, we speculated that the site (R/K/E) might be a novel key site to distinguish the conserved functions of FT, TFL and MFT (Fig. S2).

Chromosomal location, duplication and synteny analysis

We mapped the physical locations of FtPEBPs on chromosomes by using TBtools. As shown in Fig. 4, fourteen FtPEBP genes were unevenly distributed on six chromosomes (Ft1, Ft2, Ft3, Ft4, Ft5 and Ft7). Moreover, chromosome Ft4 contains the most PEBP genes (four PEBP genes), while chromosomes Ft2 has the least PEBP genes (one PEBP gene). Genome replication events have long been considered as the main driver for evolution (Ge et al., 2022). Gene duplication, tandem duplication, and significant fragment duplication tend to trigger the creation of gene families (Ge et al., 2022; Xu et al., 2012). The chromosomal region within 200 kb containing more than two homologs is defined as a tandem duplication event (Holub, 2001). Analysis of the gene duplication events of Tartary buckwheat showed that no segmental duplication occurred (Fig. S1), but there were three gene pairs (FtMFT1/2, FtFT5/6, FtTFL4/5) located in tandem repeats (Table 1, Fig. 4). These results mean most of the FtPEBP genes might evolve independently, and tandem repeat plays a significant role in FtPEBP gene family expansion.

Figure 4 Distribution of PEBP genes on Tartary buckwheat chromosomes.

The names of fourteen Tartary buckwheat PEBP genes are shown at the right side of each chromosome. Gene positions and chromosome size can be measured using the scale on the left side in mega bases (Mb). Black characters represent chromosome names and red characters represent gene names. Chromosome segments were colored in red and blue indicating high and low gene densities.

To further know the evolutionary history of PEBP genes between Tartary buckwheat and other species, collinearity analysis was performed between the genomes of Tartary buckwheat and three other plants including two model plants (Arabidopsis and rice), and a close relative of Tartary buckwheat (common buckwheat) (Fig. 5). It was found that there was only one PEBP homologous gene pair between Tartary buckwheat FtPEBP genes and Arabidopsis AtPEBP genes, three PEBP homologous gene pairs with rice OsPEBP genes and five PEBP homologous gene pairs with common buckwheat (Fig. 5). The phylogenetic tree showed that FtTFL4/5 was in the same clade with FeTFL6 of common buckwheat (Fig. 2). FtTFL4/5 has a collinear relationship with FeTFL6 (Fig. 5), but we did not detect any tandem repeat around FeTFL6 (Fig. 5). Thus, we speculated that the tandem repeat FtTFL4/5 may occur after Tartary buckwheat diverged from common buckwheat.

Figure 5 Collinearity analysis of PEBP genes between Tartary buckwheat and three other plant species.

Red lines indicate the intergenomic collinearity and red characters represent homologous genes. (A) Syntenic relationships between the homologous PEBPs of Tartary buckwheat and Arabidopsis. (B) Syntenic relationships between the homologous PEBPs of Tartary buckwheat and rice. (C) Syntenic relationships between the homologous PEBPs of Tartary buckwheat and common buckwheat.

The cis-acting element of FtPEBPs

Cis-acting elements in gene promoters have important roles in mediating transcriptional activation and repression, and numerous cis-acting elements controlling specific progresses have been reported (Hernandez-Garcia & Finer, 2014). In order to explore and understand the potential molecular function of the FtPEBP family, the 2,000 bp promoter sequences upstream of FtPEBP genes were analyzed to detect the various cis-acting elements on the PlantCARE website (Magali Lescot et al., 2002). The results suggested that many cis-acting elements were involved in the processes of light, phytohormone (auxin, abscisic acid, gibberellin, methyl-jasmonate and salicylic acid), stress (anaerobic induction, drought-inducibility, defense and stress and low-temperature responsiveness) (Fig. 6), these findings are similar with that in several other plants (Zhong et al., 2022; Zhang et al., 2023). Of these cis-acting elements, G-box, ABRE, and ARE take the most proportions among light, phytohormone, and stress responsive elements. ABRE was the most abundant element distributed in all PEBP promoters, except for the promoter of FtTFL1 (Fig. 6C). Some cis-acting elements showed gene-specific distribution patterns. More Abscisic and responsive elements (ABREs) were presented in the promoters of FtFT3, FtFT6, FtTFL2, and TFL6 (Figs. 6A, 6B), indicating these four genes might related to ABA signaling. Low-temperature responsive elements (LTRs) were mainly distributed in the FT-like subfamily. In contrast, the elements of the MYB binding site involved in drought-inducibility (MBS) were mainly detected in the MFT-like subfamily (Fig. 6). In addition, we noticed that the cis-acting elements composition of FtTFL4 are similar to FtTFL5 for their similar location in the genome, which may result from the tandem repeat. These findings revealed that the FtPEBPs could respond to light, hormones and stress to affect the development of Tartary buckwheat.

Figure 6 Regulatory elements in the promoter regions of FtPEBP genes.

(A) The number of cis-acting elements in FtPEBP promoter region. (B) The cis-acting elements distributions in FtPEBP promoters. (C) The pie charts showed the proportion of each cis-acting elements of light, phytohormone and stress response elements.

Expression analysis of FtPEBPs during the floral transition of Tartary buckwheat

To investigate the relationship between PEBP genes with the flowering time of Tartary buckwheat, we tested the expression levels of FtPEBPs in two cultivars with varied flowering time. Compared with the cultivar KQ, MQ-a rice Tartary buckwheat had a later flowering time (Fig. 7). However, they are nearly at the same growth stage because the true leaf numbers were about twelve (Fig. 7). As the flowering genes are usually expressed in leaf and floral organs to activate downstream signal cascade, we detect the expression of FtPEBP genes in leaf and inflorescence at a floral transition time in those cultivars. Among the fourteen genes, three were detected in either leaf or inflorescence tissues (Fig. 8). As shown in Fig. 8, FtFT1 had the most abundant expression level in the leaf and inflorescence of KQ, whereas it was almost not detected in late-flowering MQ. The expression level of FtFT3 was higher in the leaf and inflorescence of KQ than in MQ. The expressions of FtTFL1 were similar in both samples of all cultivars. FtFT1/FtFT3 were expressed more strongly in KQ (the early-flowering type cultivar) than in late-flowering MQ. Therefore, we speculated that FtFT1/FtFT3 might be the florigen-encoding genes positively controlling floral transition in Tartary buckwheat.

Figure 7 Different flowering time type Tartary buckwheat at 55 days after sowing and statistics of true leaf numbers at sample-harvesting time.

Figure 8 Real-time PCR analysis of FtPEBPs in the inflorescence and leaf of two Tartary buckwheat cultivars with different flowering time.

Discussion

PEBP genes play essential roles in regulating flowering time, inflorescence morphology and the formation of tubers (Karlgren et al., 2011; Susila & Purwestri, 2023; Putterill & Varkonyi-Gasic, 2016; Guo et al., 2014). The PEBP gene family has been isolated and identified from many plants, such as A. thaliana (six members) (Hedman, Kallman & Lagercrantz, 2009; Carmona, Calonje & Martinez-Zapater, 2007), O. sativa (nineteen members) (Chardon & Damerval, 2005), and Solanum lycopersicum (twelve members) (Sun et al., 2023c). Gene family is a group of genes originating from the same ancestor, produced two or more copies of one gene through gene duplication, and they are similar in gene structure and function (Xu et al., 2012). In this study, a total of fourteen FtPEBP genes were identified from the Tartary buckwheat genome by bioinformatics methods. We found that the exon-intron and motif structure were comparable among those PEBP genes. Collinearity analysis between FtPEBPs in the Tartary buckwheat genome showed no segmental repeated events in FtPEBP genes, indicating that the FtPEBPs might evolve independently. Phylogenetic analysis of fourteen FtPEBP genes was performed with model plants (Arabidopsis and rice) and common buckwheat, a related species of Tartary buckwheat. In the evolutionary relationship, one pair of homologous genes was found between Tartary buckwheat and Arabidopsis, and three pairs of homologous genes were found between Tartary buckwheat and rice. In contrast, the most homologous gene pairs (five) were found between Tartary buckwheat and common buckwheat. We speculated that this may be due to the closest relationship between Tartary buckwheat and common buckwheat.

Cis-acting elements in the promoter region often regulate gene expression. By analyzing the cis-acting elements in the promoter region of the FtPEBP genes of Tartary buckwheat, it was found that all fourteen FtPEBP promoters contained light-responsive elements, which was consistent with the previous research conclusion that photoperiod is involved in the regulation of FT and TFL1 (Wanhui et al., 2013; Pearce et al., 2017). ABRE elements are widely found in each FtPEBP, and some gene promoter regions also contain other hormone elements, such as auxin, methyl-jasmonate, salicylic acid, and gibberellin. These results indicated that the FtPEBP genes may be involved in the growth and development of Tartary buckwheat. LTR elements mainly exist in the FT-like subfamily, while MBS elements mainly exist in the TFL1-like subfamily, indicating the diverse functions between FT- and TFL-like subfamilies. The spatiotemporal-specific expression of genes may suggest the specific regulatory roles in the development of plants (Sonawane et al., 2017). In the present study, only three FtPEBPs out of fourteen genes were expressed in leaf and inflorescence. FtPEBP genes were differentially expressed in different flowering types of Tartary buckwheat. FtFT1 was only expressed in the inflorescence and leaf of early-flowering KQ. FtFT3 was more enriched in the leaf and inflorescence of early-flowering type KQ, while it was expressed at very low levels in late-flowering type MQ. The correlation between the expression levels of FtFT1/FtFT3 and the flowering time of buckwheat suggests they may be the candidate florigen-encoding genes in Tartary buckwheat. Therefore, we think FtFT1/FtFT3 could be used for yield improvement, especially for rice Tartary buckwheat, by molecular breeding approaches in the future.

Conclusions

In this study, we identified and comprehensively analyzed fourteen putative FtPEBP genes. The evolutionary relationships, gene structure and gene duplication among FtPEBPs were performed. The correlations between FtPEBP gene expression levels and the flowering time of early- and late-flowering cultivars indicates that FtFT1/FtFT3 may be involved in Tartary buckwheat’s flowering time and yield regulation. This study lays a foundation for further elucidating the potential roles of FtPEBP genes in Tartary buckwheat.

Supplemental Information

Supplemental Information 1 Synteny analysis of FtPEBPs in Tartary buckwheat genome.

Supplemental Information 2 Multiple sequence alignment of PEBP protiens.

The red arrow indicated the key amino acids distinguishing FT-like (Y), TFL1-like (H), and MFT-like (W) functions. The blue arrow indicated the other key amino acids distinguishing FT-like (R), TFL1-like (K), and MFT-like (E) functions.

Supplemental Information 3 The qRT-PCR primers used in this study.

Supplemental Information 4 The additional information for real-time PCR.

Supplemental Information 5 The raw data of various PEBP protein sequences used for phylogenetic tree analysis in this study.

Supplemental Information 6 Raw data for collinearity analysis.

Supplemental Information 7 Raw data for cis-element position of FtPEBP gene promotors.

Supplemental Information 8 Raw data for promotor sequences of various FtPEBP genes.

Supplemental Information 9 Raw data for real-time PCR.

Additional Information and Declarations

Competing Interests

Author Contributions

Data Availability

The authors declare that they have no competing interests.

Mengping Nie performed the experiments, analyzed the data, prepared figures and/or tables, authored or reviewed drafts of the article, and approved the final draft.

Li Li performed the experiments, analyzed the data, prepared figures and/or tables, authored or reviewed drafts of the article, and approved the final draft.

Cailin He performed the experiments, analyzed the data, prepared figures and/or tables, authored or reviewed drafts of the article, and approved the final draft.

Jing Lu performed the experiments, analyzed the data, prepared figures and/or tables, authored or reviewed drafts of the article, and approved the final draft.

Huihui Guo performed the experiments, analyzed the data, prepared figures and/or tables, authored or reviewed drafts of the article, and approved the final draft.

Xiao’an Li performed the experiments, analyzed the data, prepared figures and/or tables, authored or reviewed drafts of the article, and approved the final draft.

Mi Jiang performed the experiments, analyzed the data, prepared figures and/or tables, and approved the final draft.

Ruiling Zhan performed the experiments, analyzed the data, prepared figures and/or tables, and approved the final draft.

Wenjun Sun performed the experiments, analyzed the data, prepared figures and/or tables, and approved the final draft.

Junjie Yin performed the experiments, analyzed the data, prepared figures and/or tables, and approved the final draft.

Qi Wu conceived and designed the experiments, analyzed the data, prepared figures and/or tables, authored or reviewed drafts of the article, and approved the final draft.

The following information was supplied regarding data availability:

The raw data are available in the Supplemental files

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
