# Peer review of "Genome-wide identification, subcellular localization, and expression analysis of the phosphatidyl ethanolamine-binding protein family reveals the candidates involved in flowering and yield regulation of Tartary buckwheat (Fagopyrum tataricum)"

_PeerJ, doi:10.7717/peerj.17183_

## Round 0.1 · original submission · Major Revisions

The reviewers have a number of important concerns, both technical and regarding the overall design of the study. The most important one is the possible difference in the development rate between the cultivars (mentioned by two reviewers) that may obscure the interpretation of the differential gene expression analysis. There is also lack of comparison with similar, published results (rev. 1) and analysis of statistical significance (rev. 2), and many more technical points regarding both the computational and experimental paragraphs. Still, all reviewers find the overall study interesting.

·

Basic reporting

Overall, the writing style and presentation of the material are acceptable. However, several issues need fixing.
First of all, the citation database needs validation. For example, in line 141, there is a reference to the paper describing the CLASTALW tool, the citation listed as [Julie D Thompson 2002]. However, it is not a single-author paper (Thompson JD, Gibson TJ, Higgins DG. Multiple sequence alignment using ClustalW and ClustalX. Curr Protoc Bioinformatics. 2002 Aug;Chapter 2:Unit 2.3. doi: 10.1002/0471250953.bi0203s00. PMID: 18792934.). It appears that the authors were not correctly added to the citation manager.

In lines 164-169, three cultivars are mentioned. Probably, it is appropriate to add a citation to earlier papers describing the properties of these cultivars, for example, https://www.nature.com/articles/s41598-022-16001-z.

Academic English standards. According to the Chicago Manual of Style, one needs to spell out numbers zero through one hundred. For example, in the current paper, the authors need to write "three cultivars" instead of "3 cultivars".

Experimental design

The authors presented a standard computational analysis of a selected gene family. They have supplemented the in silico approach with wet-lab validation of subcellular localization analysis using GFP. Probably, the authors need to explain why tobacco leaves were used for the subcellular localization experiment. Probably, an explanation along the lines that "due to the lack of a stable genetic transformation system and effective transient expression system, [...] gene functions can only be studied via Arabidopsis thaliana, and subcellular localization can only be investigated through the heterologous expression systems, such as tobacco" (https://doi.org/10.1016/j.scienta.2022.110927) would be helpful.

Validity of the findings

My major concern is about the starting point of the study. A glance at the 2017 FT genome (https://www.mbkbase.org/Pinku1/) Pfam and Interpro annotation files shows that 17 candidate PEBP family genes have already been identified. It would be logical to at least compare this annotation to an earlier annotation. The authors have identified 15 candidates. Would be interesting to see the intersection between the previously identified PEBP and the current list.

·

Basic reporting

The manuscript is well-written with clear and unambiguous professional English throughout. The inclusion of literature references provides a solid field background and context for the study. The references are appropriate and contribute to the overall comprehensibility of the research.

Raw data, figures and table are properly used to illustrate the observations, however figures notations could be improved to include more details and explanations. For example, fig.4 has some colors that are not mentioned in the legend.

Experimental design

While the experimental setup is clearly stated and the methods are generally described with sufficient detail for reproducibility, I would like to suggest clarification on the following points:

1. How the clustering for the fig.2 was performed and evaluated? MFT-like cluster has strong bootstrap support unlike the separation of the other two branches. The figure could also be more informative in case authors kindly provide species annotation to the provided genes.

2. How the results of confocal microscopy were analyzed and interpreted? It is not clear if the fluorescence appears in cytoplasm or on the membrane. Experiment would benefit from introducing of the additional markers that have known cellular localization.

3. The samples from different cultivars were collected after the same amount of time. Since they have different developmental pace, specific developmental stage at the time of sample collection is different as well. In that case it is not clear for me, what specifically is measured in the differential expression analysis. Difference could come from both factors.

Validity of the findings

While the manuscript presents a variety of analyses, there is a notable absence of statistical significance evaluations. For instance, in the section about cis-regulating motifs near the genes of interest, motifs may have arisen by chance rather than indicating a meaningful association. It is recommended that the statistical significance of such findings be thoroughly assessed to strengthen the validity of the results and conclusions.

Additional comments

1. FtFT7 is a clear outlier in terms of gene similarity (fig. 3), could it be a false positive finding due to the misannotation?
2. What is the biological meaning of the observed motifs (fig. 3)? Do they represent functional domains or protein structure elements?
3. I would suggest to cite the corresponding papers directly instead of providing the links to the web interface. For example, for PFAM database.
4. Space after the bracket is missing on the line 142.

·

Basic reporting

In general, the manuscript is rather clear and well written and illustrated. However to my opinion it misses some important points (see below). Also, there are errors, for example: line 74: "eukaryotic community, spanning all taxa of bacteria" - this is nonsense, bacteria are not eukaryotes, lane 140 - change CLASTAL to CLUSTAL

Experimental design

I have several questions and suggestions here.
lanes 124-125: Did you use the blast search against predicted set of F.tat. proteins or against the whole genome? The second is preferrable because some genes could be missing in the annotation.
You included the data from Fagopyrum esculentum in you data set, which exactly version of F. esc. genome and annotation did you use? There is a plenty of them.
Lanes 171 - 172. You state earlier that you study three cultivars that drastically differ in flowering time. However the date of collection is the same - does not it mean that you compare samples being in different stages of development? The Figure 7 also shows this - you collected fully developed and open flowers for XQ and small floral buds for MQ. This, unfortunately, downplays your conclusions based on expression profiles - because all the differences that you observe might be due to the different stages of development, not to the differential regulation.
Also, I think this is in general the wrong choice of the sample - you should have tested earlier stages, before the flowers become visible. Since FT promotes flowering and TFL1 represses it, it is logical to expect that they act mainly before the flowering occurs (and this is true at least for TFL1 in Arabidopsis - it is expressed at early stages in the SAM/IM).
Also, there is a solid body of evidence on the functionally important amino acids that specify the activity of TFL1/FT-like protein as a repressor or activator. As Hanzawa et al. (2005) showed, a single amino acid change is sufficient to change the specificity: https://www.pnas.org/doi/10.1073/pnas.0500932102 . The same topic of the transitions from inducer to activator was developed in their further study: https://www.cell.com/action/showPdf?pii=S1674-2052%2815%2900094-5
I suggest you to expand your computational analysis by looking for those important amino acids in F.tat sequences that you identified - this might give a clue about which genes correspond to FT and which to TFL1.

Validity of the findings

I have major concerns about the conclusions based on expression analysis (see above). The ones based on computational analyses are reliable but would greatly benefit from the expansion.

Additional comments

I suggest redoing or removing the part on expression analysis and expanding the part concerning functionally important amino acids.
Also, since you mention that rice (easy dehulling) tartary buckwheat has delayed flowering time, it is interesting to look whether any candidate FT/TFL1 genes are linked with the dehulling locus. Though the causative gene is not identified, there is the information about the approximate location: https://www.ncbi.nlm.nih.gov/pmc/articles/PMC7231119/ , https://bmcgenomics.biomedcentral.com/articles/10.1186/s12864-021-07449-w
If found, this linkage could be a strong argument in favor of the hypotheses on a function of a particular FT/TFL1-like gene.

---

## Round 0.2 · Minor Revisions

There are some remaining, minor comments of the editorial character; one of the reviewers has also made comments in the manuscript. These should be taken into account.

·

Basic reporting

The authors have significantly improved the manuscript, addressing most of my concerns. However, some of my general suggestions were applied only to the part of the manuscript. To help the authors, I have prepared an annotated version of the manuscript containing all suggestions. I hope that it will be helpful for the authors.

Experimental design

No comment

Validity of the findings

No comment

Additional comments

No comment

·

Basic reporting

Manuscript was thoroughly corrected according to the reviewers comments, question were carefully addressed and corresponding adjustments were made. Now figures annotations are more clear and helpful.

Experimental design

Although there is a concern regarding the comparability of developmental stages, authors made several corrections that increase the validity of the findings.

Validity of the findings

Necessary adjustments were implemented.

·

Basic reporting

no comment

Experimental design

the authors have excluded the most early flowering cultivar so the experimental design has much improved

Validity of the findings

no comment

Additional comments

The authors have adequately revised the manuscript in line with my (and other reviewers) comments thus it can be published, with just a few minor points that should be corrected:
- no reference to Hu et al. 2023 (cited on line 283). And actually, I don't see a point in citing it in this context, this is Hanzawa et al. 2005 study that should be cited
- the raw sequences contain one sequence - FtTFL12 - that is clearly longer than other family members, it has 686 amino acid. This is probably a mis-annotation, please check and correct.

---

## Round 0.3 · accepted · Accept

The remaining editorial concerns have been addressed.